# Melatonin Promotes Uterine and Placental Health: Potential Molecular Mechanisms

**DOI:** 10.3390/ijms21010300

**Published:** 2019-12-31

**Authors:** Luiz Gustavo de Almeida Chuffa, Luiz Antonio Lupi, Maira Smaniotto Cucielo, Henrique Spaulonci Silveira, Russel J. Reiter, Fábio Rodrigues Ferreira Seiva

**Affiliations:** 1Department of Anatomy-IBB/UNESP, Institute of Biosciences of Botucatu, Univ. Estadual Paulista, Botucatu, São Paulo 18618-689, Brazil; luiz.lupi@unesp.br (L.A.L.); maira.cucielo@gmail.com (M.S.C.); hspaulonci@gmail.com (H.S.S.); 2Department of Cellular and Structural Biology, UT Health, San Antonio, TX 78229, USA; REITER@uthscsa.edu; 3Department of Biology and Technology—UENP/CLM—Universidade Estadual do Norte do Paraná, Bandeirantes, Paraná 86360-000, Brazil; fabio.seiva@uenp.edu.br

**Keywords:** melatonin, circadian clock, decidualization, uterine receptivity, implantation, placenta, endometriosis

## Abstract

The development of the endometrium is a cyclic event tightly regulated by hormones and growth factors to coordinate the menstrual cycle while promoting a suitable microenvironment for embryo implantation during the “receptivity window”. Many women experience uterine failures that hamper the success of conception, such as endometrium thickness, endometriosis, luteal phase defects, endometrial polyps, adenomyosis, viral infection, and even endometrial cancer; most of these disturbances involve changes in endocrine components or cell damage. The emerging evidence has proven that circadian rhythm deregulation followed by low circulating melatonin is associated with low implantation rates and difficulties to maintain pregnancy. Given that melatonin is a circadian-regulating hormone also involved in the maintenance of uterine homeostasis through regulation of numerous pathways associated with uterine receptivity and gestation, the success of female reproduction may be dependent on the levels and activity of uterine and placental melatonin. Based on the fact that irregular production of maternal and placental melatonin is related to recurrent spontaneous abortion and maternal/fetal disturbances, melatonin replacement may offer an excellent opportunity to restore normal physiological function of the affected tissues. By alleviating oxidative damage in the placenta, melatonin favors nutrient transfer and improves vascular dynamics at the uterine–placental interface. This review focuses on the main in vivo and in vitro functions of melatonin on uterine physiological processes, such as decidualization and implantation, and also on the feto-maternal tissues, and reviews how exogenous melatonin functions from a mechanistic standpoint to preserve the organ health. New insights on the potential signaling pathways whereby melatonin resists preeclampsia and endometriosis are further emphasized in this review.

## 1. Introduction

### 1.1. Well-Functioning Endometrium Is Essential to Ensure Embryo Implantation: General Aspects

The success of implantation relies on optimal cross-talk between embryo development and endometrial receptivity generally occurring between 6 and 10 days after ovulation. For endometrial preparation, the orchestrated control of estradiol (E2) and progesterone (P4) produced by the ovaries induces proliferative changes and development of a secretory phase, both required for embryo–endometrial synchronicity [1]. Upon P4 stimulation, the process of decidualization begins with a complex and coordinated set of molecular events that result in endometrial receptivity [2]. During decidualization, P4 influences the secretion of bone morphogenetic protein (BMP)-2 by the endometrial stromal cells to proliferate and differentiate into epithelioid decidual cells that are critical to the establishment of fetal–maternal interface [3]. There are a number of critical processes involved in endometrial development and receptivity. For instance, the homeobox (*Hox*) genes, represented by 39 *Hox* genes, encodes proteins that act functionally as transcriptional factors for the development and receptivity of the endometrium, especially during the proliferative phase with a peak in the mid-secretory phase of the menstrual cycle under P4 stimulation [4,5]; homeobox proteins (HOXA)-10 and HOXA-11 are the most important proteins that regulate other downstream factors associated with endometrial receptivity by interacting with *β*3-integrin and *Emx2* genes [6].

After the blastocyst (~0.2 mm diameter) attaches to the endometrium in a region of increased pinopode expression, a complex cascade of cytokines and chemokines and other factors such as morphogens, adhesion molecules, hormones, and transcriptional and growth factors take place to facilitate successful embryo implantation [7]. Of note, macrophages produce cytokines [e.g., leukocyte inhibitory factor (LIF) and interleukin (IL)-11 essential for embryo implantation] [8,9]. Particularly, IL-11 mediates trophoblast invasion and its deficiency is related to the reduction in endometrial natural killer (NK) cells. In the window of implantation, NK cells are the most representative immune cells, being critical regulators of angiogenesis, immunotolerance, and trophoblast migration/invasion [10,11].

Regulation of uterine peristalsis (UP) is also a relevant factor for conception and embryo implantation since UP disorders are associated with low fertility rate and may contribute to the pathogenesis of adenomyosis and endometriosis in addition to failure in sperm and embryo transport [2,12]; these UP-associated pathologies are observed in normal and infertile women and influence pregnancy in both natural and artificial cycles.

### 1.2. The Uterine Biological Rhythm as a Target for Melatonin’s Action

Every living organism possesses a circadian clock (~24 h) to execute their regular daily cycle that dictates their behaviors and physiological functions like metabolic activity, sleep/wake cycle, food intake, body temperature, and others. The master central pacemaker is the suprachiasmatic nucleus (SCN), located in the hypothalamus, which regulates photoperiodic programming of daily circadian clock [13]. SCN further coordinates the clock machinery in peripheral tissues such as lung, heart, kidney, pancreas, non-pregnant uterus, and others [14,15]. Basically, the cellular clock oscillates depending on day timing through autoregulatory transcriptional/translational feedback loops in which the heterodimer BMAL1/CLOCK drives the expression of Period (*Per*) and Cryptochrome (*Cry*) genes; after PER and CRY proteins are produced, they form dimers to suppress the transcription of *Bmal1* and *Clock* genes [14]. Furthermore, an additional short feedback loop involves the participation of BMAL1/CLOCK heterodimer on the expression *Rev-erbα* and *Rorα*, resulting in repression and activation of the *Bmal1*, respectively, through binding to nuclear orphan receptor (ROR) elements at the promoter site [16].

The uterine microenvironment responds to circadian rhythms to adapt to certain physiological functions, and fetuses are continuously exposed to hormonal and nutritional signals during gestation [17]. Akiyama and colleagues [18] observed that only maternal-derived tissues, i.e., uterus and decidua, can provide circadian information to the fetus. By using a *Period1-luciferase* (*Per1-luc*)*,* they investigated the rat uterine clock on embryonic day (E12) and on the end of pregnancy (E22) in a light–dark (LD) cycle and in constant darkness (DD). Notably, in vitro *Per1-luc* rhythms were detected in the pregnant and non-pregnant uterus with peaks corresponding to dusk in both LD and DD. Later, embryonic circadian system was proven to be self-sustained by the fetus even in the absence of maternal or environmental time cues [19]. Loss of *Per2* expression by siRNA knockdown perturbed circadian oscillations in decidualizing human endometrial stromal cells; this inhibition seems to affect mitotic expansion by blocking G2/M phase, which is positively associated with miscarriages in women with reproductive failure [20]. The involvement of REV-ERB*α* was also implicated in the molecular control of decidualization. Evidence supports an inverse role of REV-ERB*α* on bone morphogenetic proteins (BMPs) in rat uterus endometrial stromal cells (UESCs); in *Bmal1* siRNA-transfected UESCs, the *Rev-erbα* was downregulated and, consequently, Bmp2, Bmp4, and Bmp6 genes were upregulated [21]. The same investigators also observed that growth/differentiation factor (Gdf) 10 and 15 and prostaglandin G/H synthase 2 (Ptgs2), which are significantly increased during decidualization, were only upregulated after *Rev-erbα* was downregulated. Interestingly, Gdf10 and Gdf15 in addition to Ptgs2 and prostaglandin (PG)E_2_ were enhanced after UESCs were exposed to a REV-ERB*α* antagonist regardless of P4 stimulation [22,23].

As melatonin is rhythmically produced by the pineal gland during the darkness (~24 h oscillation), its output signal works by regulating and adjusting the biological clock of the cells [24], thereby inducing immediate effects and priming the day response through prospective actions during daytime [25]. The mechanism of melatonin production is governed by the rhythmic expression of *Bmal1*, *Per2*, *Cry2*, and *Rev-Erbα* in the presence of norepinephrine (NE), responsible for the upregulation of arylalkylamine-N-acetyltransferase (AANAT), one of the enzymes responsible for melatonin synthesis [26]. Of note, there are other biochemical apparatus described especially in the peripheral tissues (e.g., skin) that are capable of acetylating serotonin to N-acetylserotonin as the precursor of melatonin synthesis [27,28].

A plethora of recent studies has documented that melatonin synchronizes the clockwork machinery in healthy and damaged cells, either up or downregulating specific clock genes to maintain optimal physiology [29,30,31]. Through its multiple actions, melatonin works as a circadian regulator, natural antioxidant, and anti-inflammatory and oncostatic agent, like many other functions [32,33,34]. These effects may be receptor-dependent or independent. Melatonin receptors are formed by seven transmembrane domains coupled to high-affinity G protein, termed melatonin receptor 1 (MT1) (also called MTNR1A or Mel1a), MT2 (also called MTNR1B or Mel1b), and MT3 (identified as quinone reductase II enzyme); they often act as homo or heterodimers to reduce the levels of cAMP and cGMP thereby increasing phosphorylation of other proteins [35]. Due to its amphiphilicity, melatonin crosses the cell membrane thus interacting with intracellular molecules via different signaling pathways; these include scavenger functions, binding to calmodulin, and many other molecules [31]. The statement that melatonin binds to ROR*α* and ROR*γ* has been previously investigated and they found a lack of effect of melatonin and its derivatives on RORα and ROR*γ* activities, thereby showing by docking analyses that these nuclear receptors have no high-affinity for melatonin [36,37]. Because melatonin is rapidly metabolized in the peripheral tissues, it is worth mentioning that some of the effects may be secondary to the action of its main metabolites (e.g., 6-hydroxymelatonin, 2-hydroxymelatonin, N(1)-acetyl-N(2)-formyl-5-methoxykynuramine, and N^1^-acetyl-5-methoxykynuramine) [38,39].

Regarding the reproductive organs, clinical trials are warranted in terms of protection against the pathophysiological processes. Melatonin has long been successfully used as an efficient agent to improve ovulation and promote embryo implantation [40,41]. Moreover, it is helpful in overcoming deficiencies in fertilization and pregnancy during in vitro fertilization [40] and to protect endometrial cells from apoptosis in animals exposed to constant light [42]. Reiter and colleagues [43] reported in an extensive review the well-recognized effects of melatonin in ovary and placenta where it protects cells against oxidative stress. Likewise, maternal melatonin programs the developing master oscillator of the fetus, and a disturbed rhythm of melatonin secretion (e.g., shift work, light-at-night) generates negative consequences in the newborn (e.g., neurobehavioral problems). After the melatonin rhythm is abolished in the mother, significant changes in the expression of fetal clock genes take place; these alterations can be reversed by daily injections of melatonin [44]. During gestation, peripheral circadian oscillators may be affected by different maternal signals (e.g., nutrients, hormones, melatonin levels), which potentially result in cellular deficits in adulthood [45].

The melatonin receptors 1A and 1B have been recently shown to be expressed in human eutopic endometrium, endometriomas, and in peritoneal lesions by Mosher et al. [46]. While mRNA expression of melatonin receptors was higher in peritoneal lesions, the subtype 1A was reduced in peritoneal lesion compared with the eutopic endometrium; treatment with melatonin (dose ranging from 0.1 nM to 1.0 µM) inhibited the proliferation of endometrial epithelial cells in the presence of E2 at 48 h after exposure. It has been proposed that melatonin receptor 1B (hMTNR1B) synergizes with oxytocin to promote nocturnal uterine contractions [47]; in late-term pregnancy, circulating melatonin is of fundamental importance to induce timing and degree of contractions, and conversely, its acute inhibition with light suppresses myometrial contractions [48]. During pregnancy, hMTNR1B is suppressed but later it promotes contractile functions together with the oxytocin receptor. This internal signaling increases connexin 43 mRNA and cell to cell coupling via MTNR1B in a protein kinase C (PKC)-dependent mechanism. To confirm whether hMTNR1B is under circadian control, both immortalized human myometrial cells and primary myometrial cells were tested for BMAL1/CLOCK complex activation; overexpression of the complex led to the expression of hMTNR1B in a phase rhythm with E-box driven hPER2 [49].

### 1.3. Melatonin Actively Participates during Decidualization and Implantation Process

Melatonin has long been known to regulate uterine functions in the rat [50,51,52]. Using autoradiography, 2[^125^I] iodomelatonin binding sites were localized in antimesometrial endometrial cells while in situ hybridization with MT1 receptor cDNA probe showed expression of mt_1_ mRNAs in stromal cells [51]; dose-dependent assays with melatonin caused inhibition of forskolin-stimulated cAMP accumulation and further inhibition of [^3^H]thymidine incorporation into stromal cells, thereby revealing the antiproliferative actions of melatonin. Because decidualization progresses during pregnancy, melatonin-binding sites seem to be only active in non-decidualized outer stroma [50]. To verify the influence of E2 and P4 on the regulation of uterine melatonin receptors, ovariectomized rats displaying 70% reduction in melatonin binding sites were treated with E2 or P4 alone or in combination for 11 days; ovarian hormones induced a partial restoration of 2[^125^I] iodomelatonin binding sites followed by its MT1 receptors.

Pinealectomy reportedly increases E2 synthesis and interferes with LH and FSH release in rats [53] as a result of persistent estrus, i.e., anovulation. Levels of P4, which is needed for the development of endometrial pinopodes and microvilli [54], are also reduced after pinealectomy, a possible effect that may contribute to a reduced number of implantations. To revert this deleterious effect, daily doses of 200 µg of melatonin for 3 months restored the optimal implantation rate in adult rats and changed some uterine morphological aspects. These included a reduction in endometrial and epithelial thickness, in association with reduced number and area of uterine glands and vessels showing a weaker proliferative effect [52]. Figure 1 highlights some positive effects of melatonin in regulating decidualization and uterine receptivity.

The *p53*^−/−^ female mice exhibit failure of implantation, a reduced pregnancy rate, and smaller litter size [55]. In fact, p53 acts as the downstream signal to regulate gene encoding leukemia inhibitory factor (LIF) in uteri [56], a critical factor involved in implantation. After administering LIF to these animals, maternal reproduction was restored [55]. Notably, melatonin treatment every 12 h at increasing doses of 0.15, 1.5, 15, or 75 mg/kg body weight until day 6 of gestation promoted upregulation of progesterone receptor A (PRA), which favors the decidualization process and endometrial luminal epithelial differentiation via P4 activation. By binding specifically to MT2 receptors, melatonin significantly increased the levels of p53 to modulate LIF and facilitate implantation [57]. Although additional studies using MT1 and MT2 transgenic animal knockout models are necessary, this improvement in litter size might be of significant value to human reproductive care when considering melatonin replacement protocols. Because the majority of studies on implantation rate and melatonin are performed in animal models and cell lines (mostly using melatonin at pharmacological levels), a careful recognition of the effective dose of melatonin to be considered for clinical trials should be undertaken.

Under normal physiological conditions, the balance between oxidation/antioxidation is naturally adjusted and melatonin did not seem to exhibit uterine antioxidant properties or apoptotic effects; in this case of superoxide dismutase (SOD)1, Gpx-1, caspase-3, and Bcl-2 that were unchanged after melatonin treatment [57]. A study using 34 patients with recurrent spontaneous abortion revealed differences in proliferation/apoptosis ratio in trophoblasts and decidual cells [58]. Notably, qPCR data identified a significant increase in CDKN1A and Bax mRNAs associated with elevated levels in P53 protein; Bax is also higher in the placenta of preeclampsia compared with normal placenta. Although human studies are still lacking, melatonin has been recently shown to improve fertility while reducing uterine damage induced by torsion in 35 pregnant rats at gestational day 18 [59]. By analyzing uterine tissue on day 1 postpartum, melatonin, at dose of 10 mg/kg, significantly reduced caspase-3 and Bax and increased Bcl-2 levels. Although no changes were observed for abortion or congenital abnormalities, melatonin also reduced the expression of coiled-coil containing protein kinases 1 (ROCK1), and Hox4 and HoxD10, both involved in a number of cellular events.

Embryo quality and implantation rate are improved by melatonin in humans [60] and in experimental models [61]. Tamura et al. [60] reported an improvement in fertilization rate and oocyte quality in 56 patients undergoing in vitro fertilization and embryo transfer (IVF-ET) after treatment with melatonin (3 mg/day tablet orally at 22:00 h from the fifth day of menstrual cycle until the day of oocyte removal); authors believe that melatonin protects oocytes from free radicals while improving its quality. In addition to clinical studies, an in vitro study by Arjmand et al. [61] showed that human blastocysts encapsulated in alginate hydrogel containing decidualized endometrial stromal cells had an increased maintenance rate and survival time in the presence of melatonin; the hormones E2 and human chorionic gonadotropin were elevated at day 4 post-encapsulation, thus reinforcing the role of melatonin in developing early human embryos. After blastocysts were pretreated with melatonin and calcitonin, a higher number of implantations was observed in the endometrium of pseudo-pregnant mice; the levels of heparin binding-epidermal growth factor (HB-EGF), an important candidate molecule for facilitating endometrial receptivity and implantation, were also significantly increased in the endometrium at both mRNA and protein levels [62]. Melatonin seems to improve uterine receptivity and embryo implantation in early pregnancy by stimulating the development of the corpus luteum before gestation. A recent study by Guan et al. [63] demonstrated that melatonin given orally to mice for 14 days prior to mating was essential to improve implantation by lowering E2 secretion while upregulating the P4-regulated Indian Hedgehog (*Ihh*) factor expression in the endometrium on day 7.5 of gestation. After successful mating, melatonin treatment for 14 days increased P4 secretion and promoted enhanced expression of LIF. In these animals, although *p53*, *Hoxa11*, *COX2*, and *VEGFA* expressions were unchanged in the endometrium, the upregulation of *StAR* and *Cyp11a1* suggests P4 synthesis and promotion of the balance of P4/E2 ratio, which is crucial to favor implantation.

Melatonin has been recently introduced as an aid in promoting early mouse embryo competence for pre-implantation; as previously mentioned, this may be associated with uterine HB-EGF binding to ErbB1 and ErbB4 receptors [64]. In addition to upregulating the primary implantation receptors, ErbB1 and ErbB4, melatonin (10^−9^ M) significantly reduced intracellular ROS in mouse blastocysts while increasing total antioxidant capacity and promoting mitosis of the inner cell mass and trophectoderm cells. These effects were reversed when luzindole, a melatonin receptor antagonist, was added to the culture medium, thus suggesting these processes are melatonin receptor-dependent [64]. The implantation process depends on apposition, attachment, and penetration of the blastocyst [65]. These events are strictly regulated by *HB-EGF, p53, ErbB1*, and *COX1/2* genes and must be coordinated within the “implantation window” around day 4 of gestation. After treatment with 15 mg/kg melatonin, the mouse endometrium was thicker and developed a higher density of uterine glands compared to nontreated controls. Furthermore, melatonin decreased E2 at gestational day 6 with no alterations in P4 levels and upregulated *HB-EGF* and its receptor *ErbB1* in the blastocyst [57]. An additional experiment revealed that melatonin is beneficial for pregnant mice especially when subjected to long photoperiod (18: 6 h light: dark) exposure. In fact, long daily exposure to light had a profound impact on the number of implantation sites compared with a 12: 12 h light: dark cycle. In these animals, melatonin (10^−4^ M) may improve embryo implantation by enhancing E2 levels during pregnancy while upregulating MT1/2-mediated *p53* expression in uterine tissue. Additionally, litter sizes were increased after melatonin treatment without affecting birth weights and weaning weights of the pups [66]. These findings introduce a caution for people exposed to light pollution during their reproductive age.

In mice and humans, P4 and E2 are involved in the receptive status of the uterus to facilitate implantation [67]. The functions of E2 and P4 are mostly mediated via its receptors since mice lacking estrogen receptor alpha (ER*α*) or PRA are infertile [68,69]. Following a 24-h oscillation, MT1 and MT2 are differentially expressed in pregnant and non-pregnant uteri in humans but their involvement with P4 and E2 actions is rather complicated. Melatonin seems to increase PR and reduce ER levels in the uteri of pinealectomized rats [70]; consistently, embryo implantation and serum P4 levels are reduced after pinealectomy, and daily melatonin administration restores P4 levels [52]. We previously observed that melatonin lowers E2 and downregulates ER*β* and PRB in the rat uterus during ovulation; these regulatory effects are thought to be induced by MT1 activation [71]. During the perimenopausal period, the reduction in E2 levels is associated with a temporal increase in melatonin synthesis likely due to elevated activity of AANAT [72].

### 1.4. The Beneficial Effects of Melatonin on Placental Tissue: A Link to Preeclampsia

Melatonin plays a role in the promotion of oocyte quality, embryo implantation, fetal development, and parturition [73,74,75]. By reducing oxidative/nitrosative stress, melatonin displays considerable effects in early pregnancy and during the entire gestational period [69]. There are clinically relevant studies documenting placental production of melatonin and the presence of its receptors in both cytotrophoblasts and syncytiotrophoblasts [76,77,78]; this local production indicates paracrine and autocrine events relative to placental homeostasis and fetal development.

Both AANAT and acetylserotonin O-methyltransferase (ASMT), the enzymes required for melatonin synthesis, have been identified in human cytotrophoblasts and syncytiotrophoblasts; this production probably reduces apoptosis in the villous cytotrophoblasts while preserving survival and stable turnover of the syncytiotrophoblast cells [78]. Placental “rhythm” activity is under the control of circadian clock genes that works at the transcriptional level and via translational feedback loops; disruption of these processes compromises placental activities [79] and may further affect local melatonin production. The regulatory actions whereby melatonin impacts placental functioning are summarized in Figure 2.

During malnutrition conditions, melatonin ameliorated the diastolic function of the placental vessels in addition to stimulating uterine expression of antioxidants such as peroxidases and mitochondrial Mn-SOD [80]. During early gestation, AANAT is increased whereas MT2 was expressed only at day 2 of gestation [57]; in late pregnancy, a rise in night-time melatonin levels is observed and tends to be elevated over normal night-time after 32 weeks of gestation, thereby showing a special involvement with delivery of the newborn [41]. Changes in placental circadian rhythms (e.g., night shift work) may jeopardize melatonin production with increased risk of preterm delivery, uterine growth restriction, and even preeclampsia [78,80]. To maintain placental health, melatonin induces expression and activity of catalase and SOD, prevents oxidative stress, and inhibits the expression of vascular endothelial growth factor (VEGF) [81]. Importantly, a deficiency in melatonin production is related to spontaneous abortion in cases of no additional chromosomal anomalies or uterine malfunctions [60].

In a mouse model, melatonin prevented placental malperfusion associated with intrauterine inflammation-related oxidative stress [82]. Melatonin (10 mg/kg), when injected intraperitoneally 30 min before lipopolysaccharide (LPS) administration, prevented insufficiency of uterine and umbilical artery, placental hypercoagulation, pro-inflammatory and oxidative processes, in addition to protecting against ventricular dysfunctions in fetuses exposed to LPS-induced oxidative stress. Moreover, melatonin protected uterine and placental tissues from LPS-induced maternal inflammation with a decreased incidence in preterm birth; this effect may be attributed to the downregulation of TNF-*α*, IL-1*β*, IL-6, and COX-2 in the uterus possibly via upregulating the SIRT/Nrf2 signaling pathway [83]. There is similar evidence with pregnant women displaying low levels of melatonin and placental insufficiency (PI)-associated inflammation. The PI is a condition especially associated with intrauterine growth restriction syndrome of the fetus (IUGR) manifested at 3rd trimester of gestation; in the case of PI, melatonin levels are significantly reduced and this decrease is correlated with proinflammatory activities of TNF-*α*, IL-1*β*, and IL-6 [84].

Of particular interest is the efficacy of melatonin in the treatment of preeclampsia, a serious condition affecting some pregnant women during the last trimester of gestation. In preeclampsia, maternal hypertension, proteinuria, placental dysfunction, and disturbances in angiogenic factors are often present [85]. Both melatonin levels and its receptors are depressed during severe preeclampsia [77,78,86,87], and administration of melatonin to patients significantly attenuated preeclampsia-related oxidative stress while limiting hypertension [60,78]. After investigating placentas from pregnancies complicated by preeclampsia, Lanoix et al. [77] documented that preeclamptic placenta seems to be unable to produce melatonin since AANAT is significantly inhibited in its expression and activity; moreover, the expression of MT1 and MT2 is very low in the preeclamptic placenta compared with the normotensive placenta. Subsequently, the same authors proved that melatonin-generating enzymes, i.e., AANAT and ASMT, are normally produced in normal placenta throughout the pregnancy (from week 7 to term) with optimal expression at the 3rd trimester. In line with this, MT1 is more important at the 1st trimester for promoting villous cytotrophoblast syncytialization whereas MT2 expression is often unchanged through pregnancy [88].

Melatonin has documented benefits in protecting human placental trophoblast from hypoxia/reoxygenation (H/R) by affecting autophagy and inflammation [89,90]. By modulating 5’ adenosine monophosphate-activated protein kinase (AMPK)*α*, an upstream regulator of autophagy, and the nuclear factor erythroid 2-related factor 2 (Nrf2), a factor associated with protection against oxidative injuries, melatonin increases apoptosis in altered trophoblastic cells. Moreover, melatonin and nuclear factor kB (NF-kB) inhibition resulted in increased autophagy, which promoted syncytiotrophoblast survival in H/R conditions in complicated pregnancy.

In a phase I trial, 20 women with preeclampsia had their placenta removed at the time of cesarean for removal of the fetus; thereafter, placental tissue was cultured in the presence or absence of optimal melatonin concentration. In xanthine/xanthine oxidase placental explant model, melatonin (1 mmol/L) stimulated antioxidant molecules (Nrf2, HO-1) and reduced oxidative stress (8-isoprostane) without changing the levels of anti-angiogenic factors. Furthermore, melatonin restored endothelial monolayer integrity and mitigated vascular cell adhesion induced by TNF-*α* [91]. Since no clinical or biochemical parameters were affected by melatonin, it could serve as functional therapy to extend pregnancy duration in women with preeclampsia.

Preeclampsia has been studied experimentally [92,93]. In RUPP animals, a pregnant rat model presenting reduced uterine blood flow by using abdominal aorta and ovarian artery clipping, aggravating conditions such as hypertension, proteinuria, and failure in renal functions are present. A study by Uzun et al. [92] revealed that pinealectomy intensified these effects observed in RUPP animals (e.g., reducing VEGF, TNF-*α,* and total antioxidant status and enhancing oxidative stress index and soluble fms-like tyrosine kinase 1 (sFlt-1) expression) while treatment with melatonin reduced blood pressure (via nitric oxide pathway) and placental expression of sFlt-1; these findings may help in determining not only the melatonin effects but how melatonin deficiency influences preeclampsia. To strengthen these findings, an in vitro assay using placental explants from normal term pregnancy (>38 weeks) revealed that melatonin (100–1000 µM) potentiated the antioxidant response by upregulating the expression of thioredoxin, glutamate-cysteine ligase, and NADPH: quinone acceptor oxidoreductase 1 in the placenta while reducing sFlt-1 secretion from primary trophoblast without altering the genes related to endothelial dysfunction [93]. Whether these targeted molecules are indeed altered in the presence of physiological or pharmacological levels of melatonin in pregnant women with preeclampsia requires further investigation.

Maternal endothelium is activated by a placental “toxin” responsible for changing normal adaptation of the maternal vasculature with consequences in hypertension and other signs of preeclampsia; oxidative damage is involved with the release of these factors in the maternal endothelium [94]. One of the toxic factors released by the placenta is extracellular vesicles (EVs) derived from syncytiotrophoblast fragments at 6-week gestation. Both antiphospholipid antibodies (aPL) and preeclamptic sera stimulate the production of EVs through mechanisms that may result in the production of excessive placental free-radicals [95]. To investigate the antioxidative effect of melatonin, Zhao and colleagues exposed 25 first trimester placenta (8–12 weeks of gestation) to a PL and preeclamptic sera in the presence of melatonin; importantly, protein nitrosylation and production of endothelial-activating EVs were significantly reduced after 1 μM and 10 μM melatonin treatment. The preeclamptic serum may trigger mitochondrial dysfunction in the syncytiotrophoblast. Recently, trophoblast cells collected from the first trimester placentas of ten obese women showed greater intrauterine oxidative stress. After treatment with melatonin (10 and 100 µmol/L), the maximal mitochondrial respiration and its sparing capacity were elevated in addition to an elevated expression of glutathione peroxidase [96].

### 1.5. Melatonin’s Action on the Immune Microenvironment at the Maternal–Fetal Interface: A Brief Overview

Pregnancy loss is associated with immunological disturbances and oxidative stress; in this perspective, melatonin serves as a direct free-radical scavenger possessing immunomodulatory properties [97]. The precise mechanism by which the fetus escapes immune rejection by the mother remains inconclusive; however, there is increasing evidence that associates the circadian clock and melatonin with pregnancy and synchronous programming of the immune system [92]. Melatonin is rhythmically produced by the placental cells and is related to lymphocyte proliferation and cytokine production [78,98]. Reproductive failures including recurrent abortion, loss of implantation, and preeclampsia are all related to alterations in NK cells, Tregs, Th1/Th2 ratio, and Th17 [98]. Uterine NK cells represent the majority of the lymphocyte population during the luteal phase and early gestation [99]; decidual NK cell and peripheral blood NK cells have different phenotypes of CD56^dim^ to CD56^bright^. The decidual cells signal to activate its resident immune cells and attract macrophages, monocytes, neutrophils, and NK cells [98,100]; leukocytes also enhance TNF-*α* and IL-6 levels in addition to synthesizing more prostaglandins for uterine contractions [101]. Consistent with these findings, the melatonin rhythm is synchronically correlated with Th1/Th2 ratio in the maintenance of fetal survival [102] while inducing T-lymphocyte precursors thereby affecting monocyte function and increasing NK cell activity; the elevation in IL-2, IL-6, IL-12, and IFN-*γ* by the Th cells may be responsible for functional enhancement of NK cells [103].

Macrophages are a second subset of immune cells in decidualized tissue and are mainly involved in the renovation of normal cellularity during early and late pregnancy [104]. Based on the innate immunity, in vivo and in vitro studies showed that melatonin promotes an inhibitory effect on the expression of B7-1, iNOS, and COX-2 in macrophages [105] while upregulating pro-inflammatory cytokines (e.g., TNF-*α* and IL-1) and macrophage-related activities [106]; how melatonin influences immune tolerance through the regulation of decidual macrophages still remains to be determined. 

During gestation, changes in melatonin levels have a prominent role in the activity of the immune processes in women. Of note, melatonin is thought to control the number of the T-cell subpopulations, Th17 and Tregs, both implicated in the development of gestation [107]. Both physiological and pharmacological levels of melatonin induce in vitro differentiation of intact CD4 + T cells in Th17 [108]. Recently, ex vivo and in vitro studies using auto-serum as a source of endogenous melatonin, showed differentiation in both CD4 + ROR*γ*t + and CD4 + FoxP3 + T cells, leading to an inverse balance of Th17/Treg [109]. Although melatonin’s functions on the maternal–fetal immune cells are not completely defined, exogenous melatonin is documented to have beneficial effects for mother and fetus in the context of immunocompromised pregnancy. On the contrary, if disturbances in melatonin synthesis are implicated in breaking the maternal–fetal immune tolerance remain unraveled.

## 2. The Benefits of Melatonin in the Treatment of Endometriosis

Endometriosis (EMS) is a chronic inflammatory and estrogen-dependent disorder in which the ectopic endometrium grows outside the uterine locations such as in the ovary or other pelvic tissue and in peritoneal surface; in most cases, this seems to arise by retrograde menstruation [110]. Beyond the classical pelvic pain, EMS is associated with poor quality of life, reduced work productivity, and with a number of clinical repercussions such as affective disorders, metabolic dysfunctions, anxiety, inflammation, cardiovascular diseases, and increased cancer risks [111]. The origin of EMS appears to be related to genetic and environmental factors, but its causality requires additional evidence. There are still considerations that an endocrine-disrupting chemical, refluxed tissue, presence of Mullerian rests, bone marrow-derived stem cells, and hematogenous and lymphatic dissemination may be involved in the development of EMS [112]. Numerous molecular defects are described in eutopic endometrium of women with EMS, such as oncogenic pathways, high production of estrogen and its receptor ER*β*, high levels of cytokines, prostaglandins, and metalloproteinases (MMPs) [113]. Some agents (e.g., gonadotropin-releasing hormone antagonist, cyclooxygenase inhibitors) associated with suppression of ovulatory menses and estrogen production, and surgical removals of lesions are used to control the pelvic pain [113]; however, the pathology of this complex disease remains an unsolved issue.

Circadian disruption is associated with lower levels of melatonin and a higher risk of EMS [114]; the melatonergic pathway is thought to be suppressed in EMS. Conversely, endometrial biopsies of women with endometriomas (*n* = 20) or peritoneal lesions (*n* = 11) showed variations in the expression of melatonin receptors (MR1A and MR1B) according to the site of the ectopic tissue [46]. Pinealectomy, in a rat model of EMS, caused an augment in spherical explant volume accompanied by higher lipoperoxidation and lower antioxidant activities (e.g., SOD and CAT); melatonin efficiently reversed these EMS-related parameters [115]. Melatonin has been proven to have greater effectiveness than letrozole, an aromatase inhibitor, during the treatment of EMS [116]; these effects are mostly attributed to a reduction in endometriotic foci and histopatologic scores associated with increased levels of SOD and CAT. Figure 3 depicts the molecular mechanisms by which melatonin hampers EMS progression.

To investigate melatonin’s action on endometriotic cells collected from women with EMS, a model of severe combined immunodeficient (SCID) mice were administered subcutaneously with endometriotic cells. After EMS induction, SCID animals received 20 mg/kg/day of melatonin for four weeks; implantation rates were increased after melatonin administration and the levels of malondialdehyde (MDA) were significantly reduced in the presence of E2 without changing SOD levels [117]. The same group showed that melatonin, especially at higher doses, is effective in promoting the regression of endometriotic lesions in oophorectomized rats [118]. Rats that were experimentally induced for EMS and received daily doses of melatonin had a prominent reduction of endometrial explants in addition to a decrease in COX-2 and MDA levels and enhanced activities of SOD and CAT [119]. Furthermore, when melatonin (10 mg/kg/day) was intraperitoneally injected into endometriotic cell-implanted rats, the weight of implants and related histologic scores were profoundly reduced. In addition, activities of SOD and TIMP-2 were higher after melatonin treatment in contrast to reductions in VEGF and MMP-9 levels in implanted tissues [120].

EMS is difficult to analyze during its early phase in humans. The activity of MMPs in the pathogenesis of EMS is believed to be one of the central players in the formation of endometriotic lesions [121]. To verify the involvement of melatonin in the early development of EMS, ovariectomized mice were induced with endometriotic samples of patients and implanted with E2 pellets [122]. After developing EMS, the animals were given melatonin at doses of 48 mg/kg/day for 10–20 days; notably, melatonin promoted apoptosis in endometriotic region via Bax and caspase-9 activation along with reduction in Bcl-2 expression. Melatonin also suppressed early induction of MMP-3 through regulation of c-Fos and TIMP-3 and attenuated EMS through caspase-3-mediated signaling. In an earlier study, Paul et al. [123] had reported that melatonin prevented lipid peroxidation and protein oxidation while downregulating MMP-9 activity and expression in peritoneal EMS in mice; the reduced activity of MMP-9 was negatively correlated with TIMP-1 expression, thereby indicating MMP-9/TIMP-1 ratio as a novel marker for EMS progression.

Ectopic endometriotic tissue often presents with higher mitochondria-related ERβ, which is linked to increased bioenergetics associated with ROS production and enhanced SOD2 levels [124,125]. Increased production of N-acetylserotonin (NAS), at the expense of melatonin, seems to drive cell proliferation in EMS; the NAS/melatonin ratio may further result in elevation of estrogen and mitochondria ERβ. Data arrays of the pathoetiology of EMS have determined an increase in the mitochondria N-acetylserotonin (NAS)/melatonin ratio and a decrease in the levels of vitamin A and its retinoic acid derivatives as a possible molecular link to EMS development in women [126].

In a more recent investigation of endometriotic epithelial cells, Qi et al. [127] reported that the expression of Notch1, Slug, Snail, and N-cadherin is increased whereas expression of E-cadherin and Numb is reduced in the eutopic endometrium of patients; melatonin, at doses of 1 mM, significantly inhibited E2-induced cell proliferation and attenuated migration and invasion via upregulation of Numb and low activity of Notch in these patient-derived cultured cells. The efficacy of melatonin in EMS has also been documented in a phase II double-blind clinical study; in addition to improving the sleep quality, melatonin regulated the secretion of brain-derived neurotrophic factor (BDNF) while reducing daily pain, dysuria, and dysmenorrhea by 40% [128].

It has been consistently demonstrated that endometrial immune cells play a role in the pathophysiology of EMS and in related morbidities of pelvic pain and infertility; notably, pro-inflammatory M1 macrophages are higher than the anti-inflammatory M2 phenotype, and NK cells showed abnormal function in the endometrial tissue [129]. Since the escape of endometrial tissue involves proliferation, migration, adhesion, as well as angiogenesis, melatonin is suggested to have a contributory role in the control of ectopic immune cells. Yang et al. [130] summarized the pleiotropic role of melatonin in a wide variety of gynecological disorders affecting women at reproductive age. New clinical trials mapping the melatonin levels in patients with early symptoms of EMS are urgently needed.

## 3. Concluding Remarks and Perspectives

Molecular disturbances in the circadian clock by external signals (e.g., light pollution) or by other environmental processes affecting internal body clock oscillation are profoundly associated with low rates of implantation, pregnancy deficiencies, higher incidence of menstrual irregularities, infertility, and miscarriage in women. These events are often accompanied by reduced levels of melatonin and its receptors in normal uterine tissue and in the placenta. In this context, the indole reportedly participates in numerous functions to provide a suitable decidualization/implantation process, and placentation, which may, in turn, favor the complete gestation period until parturition. The most studies evaluating decidualization and uterine receptivity are conducted with animal models treated with higher doses of melatonin, being sometimes difficult to translate to a human physiological response; whether melatonin is capable of interfering with E2 and P4 to regulate molecular targets involved in endometrial differentiation and blastocyst attachment in women remains to be determined.

To protect cells and tissues against the oxidative damage, local production of melatonin in uterine and placental compartments is extremely important; to do so, mitochondria, as the main source of its synthesis, must have their function preserved. By acting as a multitasking agent, melatonin is documented to interact with endometrial and placental cells providing antiapoptotic, antioxidative, and anti-inflammatory effects. Melatonin surely sends a signal to these cells to organize their physiological functions. Since uterine/placental activities occur periodically, the melatonin signaling may be essential to synchronize daily molecular rhythms of cells, i.e., secretion of hormones and factors, control of proliferation and apoptosis, regulation of the metabolic status and free radical scavenging, differentiation and activation of immune cells, maintenance of vascular dynamics, and others.

In addition to placental synthesis, exogenous melatonin easily crosses the human blood–placental barrier [131], potentially reaching elevated levels and being available for maternal–fetal utilization. Although the benefits of melatonin during the pregnancy is indisputable (especially during the last trimester of gestation where it improves P4 synthesis and inhibits premature oxytocin release), it has been consistently shown that melatonin works to control the pathogenesis of neonatal morbidities and diseases associated with inflammation, cell death, and oxidative stress [132]. Clinical trials have revealed the potential of melatonin as an adjunctive in the treatment of newborn, especially in term infants. However, more studies evaluating the pharmacological levels of melatonin and its presence in human reproductive tissues are of fundamental importance.

Melatonin plays a vital role in protecting female reproduction from gynecological and obstetrical pathologies (e.g., preeclampsia and endometriosis) as the circadian disruption of melatonin has been positively correlated with placental malfunction and ROS production. Experimental evidence supports a role of melatonin in providing adequate placental perfusion while preventing vascular damage, inflammation, and local oxidative stress. At pharmacologically relevant levels, melatonin reduces ROS and hypertension in preeclamptic tissue and may be considered useful as a natural adjuvant for the treatment of preeclampsia. Due to its antiproliferative, antioxidant, anti-inflammatory, and anti-invasive properties, melatonin also acts as a potential candidate to treat endometriosis; via its antiestrogenic effects, melatonin reduces the growth of E2-induced endometriotic tissue. Although most of the beneficial effects of melatonin are obtained from experimental studies using animals, there is also clinical evidence documenting the pharmacological treatment with melatonin may suppress growth and invasion of endometriotic cells through several molecular mechanisms, besides favoring sleep quality and reducing the unpleasant symptoms of patients. The understanding of endometriosis as an estrogen-related disease has led to the development of therapies involving estrogen inhibition (e.g., hormonal contraceptives and progestins), and into this perspective, melatonin may be considered a promising agent to act as an adjunctive. Of note, additional studies on women experiencing low levels of melatonin and the risk of developing endometriosis are necessary.

Besides having a contributory role in the promotion of oocyte quality, fertilization, embryo implantation, pregnancy, and parturition, melatonin has been shown to facilitate implantation rate in women undergoing assisted reproductive treatment [60,133] and in patients with polycystic ovary syndrome undergoing intrauterine insemination [134]. Considering its absence of toxicity, more attention should be given to the potential benefits of melatonin on reproduction. Furthermore, people need to be advised about the real necessity of replacement and on specific dosages for each treatment modality. It is probably best to preserve endogenous melatonin production by identifying its uptake by the uterus and placenta as an alternative to taking it as a supplement.

## Figures and Tables

**Figure 1 ijms-21-00300-f001:**
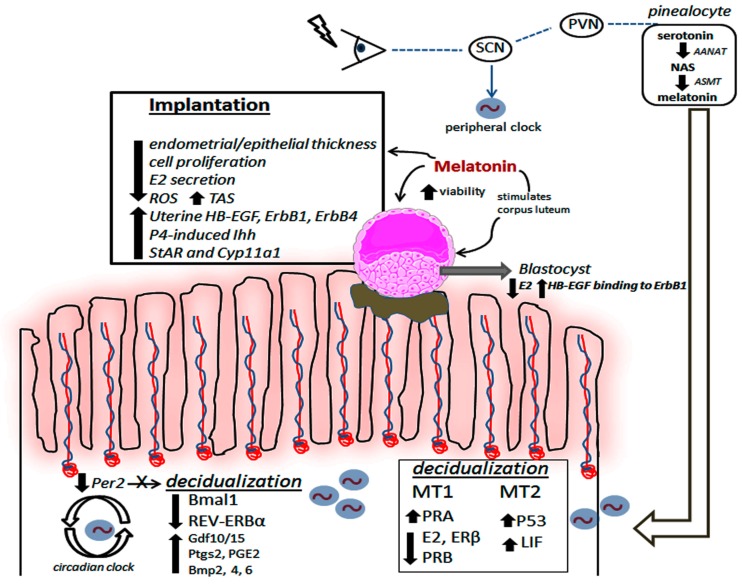
Cyclic production of melatonin, from external (e.g., pineal and possibly extrapineal sources) or internal uterine sources, is responsible for promoting decidualization and improving implantation. Circadian disruption of Per2 affects decidualization and a significant reduction in Bmal1 and REV-ERB*α* may favor the process via different molecular signaling pathways. Melatonin seems to reduce E2 while increasing the levels of decidualizing-related molecules, i.e., P53 and LIF, in the uterine tissue. In the blastocyst, melatonin directly induces higher viability (by reducing E2 and enhancing HB-EGF-ErbB1 pathway) and also indirectly by stimulating corpus luteum activity. To improve implantation, melatonin modifies the endometrial and epithelial thickness by lowering E2 levels while increasing its glandular density; these, in turn, result in ROS attenuation with high production of TAS. Importantly, melatonin significantly augments HB-EGF, ErbB1, and ErbB4 and promotes activity of P4-related factors in the uterine tissue. SCN: suprachiasmatic nucleus, PVN: paraventricular nucleus, AANAT: aralkylamine N-acetyltransferase, NAS: N-acetylserotonin, ASMT: acetylserotonin O-methyltransferase; MT1 and MT2: melatonin receptors 1 and 2; E2: 17β-estradiol, ER*β*: estrogen receptor *β*, PRA and PRB: progesterone receptors A and B; P53: tumor suppression protein p53, LIF: leukemia inhibitory factor, HB-EGF: heparin-binding EGF-like growth factor, ErbB: epidermal growth factor receptor; P4: progesterone, Ihh: Indian hedgehog signaling molecule, ROS: reactive oxygen species, TAS: total antioxidant substances, Bmal1: Brain and Muscle ARNT-Like 1; Per2: period 2, REV-ERB: nuclear receptor; Gdf: growth differentiation factor, Ptgs: prostaglandins; Bmp: bone morphogenetic protein, StAR: steroidogenic acute regulatory protein, Cyp11a1: cytochrome P450 family 11 subfamily A member 1.

**Figure 2 ijms-21-00300-f002:**
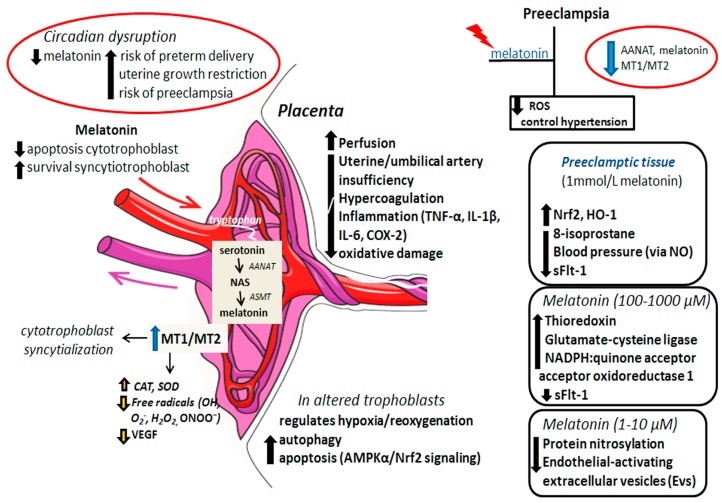
Circadian disruption with a decline in melatonin levels has been associated with placental disturbances possibly affecting maternal and fetal health. Melatonin and its receptors MT1 and MT2 are constantly synthesized by the placental cells to promote survival and syncytialization of the cytotrophoblast while reducing free radicals and angiogenesis. Through still via unknown mechanisms, melatonin enhances placental perfusion and prevents artery insufficiency, hypercoagulation, and inflammatory and oxidative damage; melatonin also protects altered placental trophoblasts from hypoxia/reoxygenation by regulating apoptosis and autophagy. In the preeclampsia condition, enzymatic machinery for melatonin synthesis is inhibited in addition to the low expression of MT1 and MT2 receptors; exogenous administration of melatonin has a prominent role in controlling ROS and hypertension. Experimentally, the treatment of preeclamptic tissues with different doses of melatonin has evidenced positive responses against various deleterious parameters; in general, melatonin attenuates ROS, restores the antioxidant status of tissue, recovers endothelial monolayer integrity and mitigates vascular cell adhesion. ROS: reactive oxygen species, AANAT: aralkylamine N-acetyltransferase, NAS: N-acetylserotonin, ASMT: acetylserotonin O-methyltransferase; MT1 and MT2: melatonin receptors 1 and 2, CAT: catalase, SOD: superoxide dismutase, VEGF: vascular endothelial growth factor, AMPK*α*: AMP-activated protein kinase *α* subunit, Nrf2: nuclear factor erythroid 2-related factor 2, HO-1: heme oxygenase-1, NO: nitric oxide, NADPH: reduced nicotinamide adenine dinucleotide phosphate, TNF-*α*: tumor necrosis factor alpha, IL-1*β*: interleukin 1*β*, IL-6: interleukin 6, COX-2: cyclooxygenase 2, sFlt-1: soluble fms-like tyrosine kinase 1.

**Figure 3 ijms-21-00300-f003:**
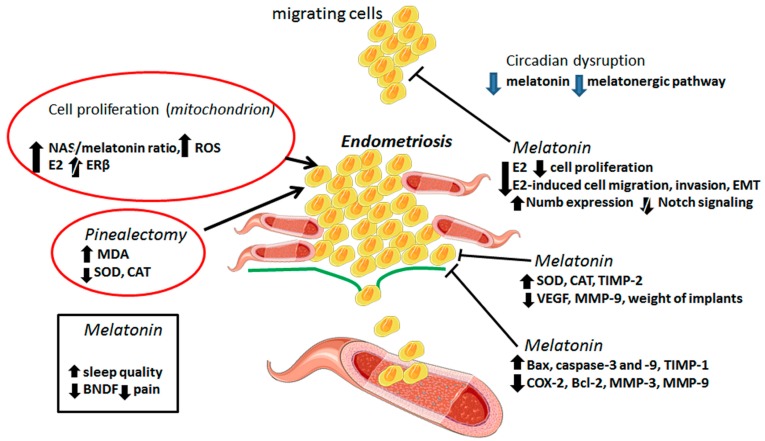
This figure illustrates possible mechanisms by which melatonin influences endometriosis (EMS). It is well-accepted that elevations in mitochondrial NAS/melatonin ratio, E2 levels, and ROS formation are associated with EMS development; other conditions like pinealectomy and circadian clock disruption are also reported to suppress melatonin which may possibly enhance the risk of EMS. Through a number of mechanisms, melatonin is documented to control growth and invasion of endometriotic cells; these include regulation in matrix remodeling, suppression of the epithelial to mesenchymal transition (EMT), induction of apoptosis, reduction of angiogenesis, and enhancement in antioxidant activity. Furthermore, melatonin promotes a reduction in the size of implants while improving sleep quality and alleviating pain or discomfort in EMS patients. NAS: N-acetylserotonin, ROS: reactive oxygen species, E2: 17*β*-estradiol, ER*β*: estrogen receptor subunit *β*, MDA: malondialdehyde, SOD: superoxide dismutase, CAT: catalase, VEGF: vascular endothelial growth factor, TIMP: metallopeptidase inhibitor, MMP: matrix metalloproteinase, Bax: BCL2 associated X, apoptosis regulator, Bcl-2: B-cell lymphoma protein 2, antiapoptotic protein, COX-2: cyclooxygenase.

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
