# Peer review of "Melatonin Promotes Uterine and Placental Health: Potential Molecular Mechanisms"

_ijms, 2019, doi:10.3390/ijms21010300_

Round 1

Reviewer 1 Report

All effects described, at laboratory level and at organism level (mostly rats), are at melatonin concentrations (mM) and dosage levels (10-24 mg/kg) far above endogenous concentrations and pharmacological doses needed for efficacy in sleep. Are findings really a specific effect of melatonin or are we looking at non specific effects?

Author Response

#Reviewer 1

All effects described, at laboratory level and at organism level (mostly rats), are at melatonin concentrations (mM) and dosage levels (10-24 mg/kg) far above endogenous concentrations and pharmacological doses needed for efficacy in sleep. Are findings really a specific effect of melatonin or are we looking at non specific effects?

Response: Thank you for considering our review article. In fact, most studies involving animal models and cells used pharmacological doses to treat female reproductive diseases (e.g., preeclampsia, endometriosis, and others). When circulating melatonin are reduced the highest the risk of developing the diseases; however, the effective results on melatonin treatment are at mM or µM (for cells) or varying between 10-24 mg/kg (for experimental models) which are considered really far physiological doses. By acting through different mechanisms (receptor-dependent or -independent), melatonin produces direct effect on scavenging ROS/RNS and indirect effects thereby regulating inflammation, angiogenesis, endometrial decidualization, like many others. There are also clinical trials reporting physiological doses of melatonin to improve symptoms of these diseases but they require further investigation to define best dose and regimen of treatment. We tried to carefully identify the in vivo and vitro effects of melatonin and even in clinical trials to make an easy correlation with dose and experimental strategy in the text (please see highlighted text over the manuscript).

Reviewer 2 Report

This is in deed a very extensive and nicely done review on the topic, and should be published in IJMS. The topic would be of broad interest, especially, looking into the role of cell extrinsic signaling pathways in deciding the endometrial physiology and health.

I have a couple of minor comments:

(1) Can the authors bring in/discuss about a bit more disease relevance in future implications. They have discussed about clinical phenotypes and how they are associated with different pathways in different sections, but it will be really useful to summarize them in the concluding section.

(2) The sentence in line 44/45 is not very clear: “During decidualization, P4 influences the secretion of bone morphogenetic protein (BMP)-2 via transforming growth factor (TGF)-β to proliferate and differentiate into epithelioid decidual cells. Please explain this a bit more. It is also a bit strange to see a parenthesis while naming signaling pathways. I would prefer not to use such a style. It is confusing.  

Author Response

#Reviewer 2

This is in deed a very extensive and nicely done review on the topic, and should be published in IJMS. The topic would be of broad interest, especially, looking into the role of cell extrinsic signaling pathways in deciding the endometrial physiology and health.

Response: Thank you for considering our review article. Your inputs will improve the quality of the review.

I have a couple of minor comments:

(1) Can the authors bring in/discuss about a bit more disease relevance in future implications. They have discussed about clinical phenotypes and how they are associated with different pathways in different sections, but it will be really useful to summarize them in the concluding section.

Response: We agree with the reviewer. Now we added more information on related diseases from a clinical standpoint (please see the sentences on pages 12 and 13).

(2) The sentence in line 44/45 is not very clear: “During decidualization, P4 influences the secretion of bone morphogenetic protein (BMP)-2 via transforming growth factor (TGF)-β to proliferate and differentiate into epithelioid decidual cells. Please explain this a bit more. It is also a bit strange to see a parenthesis while naming signaling pathways. I would prefer not to use such a style. It is confusing.  

Response: As recommended by the reviewer, we changed the sentence to avoid misinterpretation (please see highlighted text on page 2). Decidualization, the process by which the uterine endometrial stroma proliferates and differentiates into large epithelioid decidual cells, is critical to the establishment of fetal-maternal communication and the progression of implantation. This process of cell proliferation and differentiation is stimulated by P4 and BMP-2 and is mediated by TGFB. We removed TGFB pathway in order to clarify the sentence.

Reviewer 3 Report

This manuscript by Chuffa and colleagues is well-written and timely. The scope of this review is very broad, which is of course good, but also generates some problems. For example, the citations jump from a clinical study to a cell culture study to animal data so much, that often statements could be considered misleading. The authors need to be more explicit when the data are from rodents versus clinical pilot studies. Similarly - and this is an issue that happens frequently in the melatonin literature - the use of constant exposure melatonin (for example, in cell culture experiments) can not be considered equivalent to the natural nocturnal elevations of plasma melatonin. In this same regard, concentrations of 1000 micromolar melatonin are probably never seen by any tissue, so what do experiments with such concentrations have to do with physiological processes?? Here again, the authors need to be cautious about interpreting conclusive "effects" of melatonin in normal (or abnormal) states without providing convincing evidence that these pharmacological levels are present in normal tissues. Another point that needs to be made is to state the obvious - correlation does not mean causation (for example see p 10, line 434). And finally, it is well-known that the reproductive physiology of female rodents is not adequate to draw inferences about human fertility (see Mitchell & Taggart 2009 Amer J Physiol Regul Integr Comp Physiol 297: R525). So again, the authors should be cautious in drawing inferences from the rodent literature.

Author Response

#Reviewer 3

This manuscript by Chuffa and colleagues is well-written and timely. The scope of this review is very broad, which is of course good, but also generates some problems. For example, the citations jump from a clinical study to a cell culture study to animal data so much, that often statements could be considered misleading. The authors need to be more explicit when the data are from rodents versus clinical pilot studies. Similarly - and this is an issue that happens frequently in the melatonin literature - the use of constant exposure melatonin (for example, in cell culture experiments) can not be considered equivalent to the natural nocturnal elevations of plasma melatonin. In this same regard, concentrations of 1000 micromolar melatonin are probably never seen by any tissue, so what do experiments with such concentrations have to do with physiological processes?? Here again, the authors need to be cautious about interpreting conclusive "effects" of melatonin in normal (or abnormal) states without providing convincing evidence that these pharmacological levels are present in normal tissues. Another point that needs to be made is to state the obvious - correlation does not mean causation (for example see p 10, line 434). And finally, it is well-known that the reproductive physiology of female rodents is not adequate to draw inferences about human fertility (see Mitchell & Taggart 2009 Amer J Physiol Regul Integr Comp Physiol 297: R525). So again, the authors should be cautious in drawing inferences from the rodent literature.

Response: Thank you for considering the scientific value of our review article. Your inputs improved the quality of the review. This reviewer is correct. There are many studies using different protocols to test melatonin in animals and in patient-derived cells which is sometimes a bit confusing. So, we tried to elucidate (yellow highlights in the text) when the results are from rodents or culture cells versus clinical studies (we need to consider all of them to bring new information on melatonin effects – most of experimental studies are indeed with pharmacological doses of melatonin while some clinical trials used melatonin at 10-15 mg). We have to keep in mind that low physiological levels of melatonin are associated with risk of developing reproductive failures and, in some cases, pharmacological doses are the only manner to revert the condition (especially in experimental condition). To verify if the efficacy of pharmacological levels of melatonin reaching tissue concentrations are equivalent or discordant, more clinical trials are needed. As recommended by this reviewer, we were caution to state and conclude that these pharmacological concentrations are present in human reproductive tissues and we also clarified that most of studies are done in animal models and in cells collected from damaged placenta or endometriotic cells (please see conclusion section).         

Round 2

Reviewer 1 Report

Thank you for adding the texts as highlighted, to put the findings in context. 

Author Response

Thank you for your suggestions in order to improve the quality of the manuscript.